# Tangentially Active Polymers in Cylindrical Channels

José Martín-Roca and Chantal Valeriani

*Departamento de Estructura de la Materia, Física Termica y Electronica,*
*Facultad de Ciencias Físicas, Universidad Complutense de Madrid, 28040 Madrid, Spain*

Emanuele Locatelli

*Department of Physics and Astronomy, University of Padova,*
*Via Marzolo 8, I-35131 Padova, Italy and INFN,*
*Sezione di Padova, Via Marzolo 8, I-35131 Padova, Italy*

Valentino Bianco

*Faculty of Chemistry, Chemical Physics Department, Complutense University of Madrid,*
*Plaza de las Ciencias, Ciudad Universitaria, Madrid 28040, Spain*

Paolo Malgaretti

*Helmholtz Institut Erlangen-Nürnberg for Renewable Energy (IEK-11),*
*Forschungszentrum Jülich, Cauer Str. 1, 91058, Erlangen, Germany*

(Dated: May 6, 2024)

We present an analytical and computational study characterizing the structural and dynamical properties of an active filament confined in cylindrical channels. We first outline the effects of the interplay between confinement and polar self-propulsion on the conformation of the chains. We observe that the scaling of the polymer size in the channel, quantified by the end-to-end distance, shows different anomalous behaviours at different confinement and activity conditions. Interestingly, we show that the universal relation, describing the ratio between the end-to-end distance of passive polymer chains in cylindrical channels and in bulk is broken by activity. Finally, we show that the long-time diffusion coefficient under confinement can be rationalised by an analytical model, that takes into account the presence of the channel and the elongated nature of the polymer.

## I. INTRODUCTION

Active systems are characterized by the presence of mechanism that breaks equilibrium at the microscopic scale, generating directed (self-propelled) motion[1]. As a consequence, they display dynamical and collective properties that are vastly different from those displayed by their passive counterparts[2]. Of particular interest, for biological as well as for synthetic systems, is the behaviour of active matter in complex or confined environments[3]. For example, since the seminal work of Rotschild [4] we know that sperm cells accumulates at boundaries, a behavior that is common to bacteria [5, 6] and algae [7]. Such behavior has also been reported for theoretical models that capture the far-field velocity profile of microswimmers under confinement [8, 9], and for both theoretical and experimental results dealing with phoretic colloids [10]. Recently, the characterization of active filaments has become a cutting edge research in active matter [11], for two main reasons. Firstly, systems composed by active polymer-like units are ubiquitous in Nature at different lenght scales, from the sub-cellular level[12–15], to bacteria or other micro-organisms[16–19] all the way to worms and other multi-cellular organisms[20–25]. Secondly, thanks to technological progress, the synthesis of artificial active chains [26–29] and soft robots[30, 31] is now possible. Such synthetic analogues have various possible applications[32, 33].

From the modeling perspective, active polymers are macromolecules composed by out-of-equilibrium beads, whose activity can arise from a temperature mismatch[34–36] or from a self-propulsion force, completely random[37] or oriented along the polymer backbone[1, 38, 40–47]. We focus on the latter case, sometimes also referred to as polar active polymers, as they are believed to mimic the action of molecular motors[48]. Despite the growing interest in the field, few works have investigated the properties of active filaments under confinement.

Notably, the effects of spherical confinement have been considered for different models[13, 49–52]: spherical confinement is indeed relevant for biophysical systems such as chromatin. Further, the dynamics in complex confinement, such as porous media, has received some attention[53–57]. Other settings, such as translocation[58], slab[59] or cylindrical confinement[60–62] have been rather overlooked. This is surprising, given that understanding the effects of cylindrical and slab confinement has paved the way for understanding more complex scenarios, for passive polymers and active matter alike[3, 63–65].

In this paper we investigate the conformation and dynamics of tangentially active polymers under cylindrical confinement (as shown in Fig.1). We will show that, as compared to their passive counterparts, the configurational properties of polar active polymers display a rich

scenario, that stems from the interplay between confinement and activity. Importantly, these polymers do not follow the same universal curve, reported in the passive case[66]. This suggests that the blob picture, that holds for polymers in equilibrium, is no longer valid and, similarly to [60], the polymer self-similarity is broken at some non-trivial length scale. We also show that, at variance with active colloids, confinement does not always lead to accumulation to the channel walls. Rather, such accumulation appears only under weak confinement conditions. Finally, we find that the long-time diffusion coefficient along the channel axis is enhanced for long polymers with respect to the bulk value, even in the weak activity limit. By means of an analytical approximation, we show that this enhancement can be rationalized by a "rod-like" effect, that synergizes with the polar self-propulsion force and effectively increases the polymer diffusion.

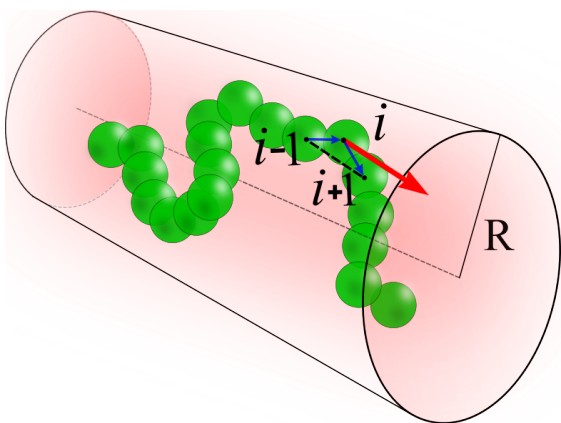

FIG. 1. Schematic sketch of the active polymer in a cylindrical channel. Polymer is constructed as beads connected by linear springs. For each bead $i$ the active force is applied in the direction of the vector $(\mathbf{r}_{i+i} - \mathbf{r}_{i-i})/|\mathbf{r}_{i+i} - \mathbf{r}_{i-i}|$.

## II. MODELS AND METHODS

### A. Numerical model and simulation details

We model active polymer chains in a coarse-grained fashion; a sketch is reported in Fig.1. We employ the well known bead-spring Kremer-Grest model[67], where each polymer consists of $N$ monomers, that interact with each others via a repulsive WCA potential

$$U^{\text{WCA}} = \begin{cases} 4\epsilon \left[ (\sigma/r)^{12} - (\sigma/r)^6 + \frac{1}{4} \right], & r < 2^{1/6}\sigma \\ 0, & \text{else} \end{cases} \quad (1)$$

where $\sigma$ is the monomer diameter and $\epsilon = 10k_BT$, $k_B$ being the Boltzmann factor and $T$ the absolute temperature. Neighbouring monomers along the backbone are held together by a FENE potential

$$U^{\text{FENE}} = \begin{cases} -0.5KR_0^2 \ln \left[ 1 - (r/R_0)^2 \right], & r \leq R_0 \\ \infty, & \text{else} \end{cases} \quad (2)$$

where we set $K = 30\epsilon/\sigma^2 = 300k_BT/\sigma^2$ and $R_0 = 1.5\sigma$. This choice of parameters allows to avoid strand crossings. We introduce the polymer's activity in the form of a tangential self-propulsion force. All monomers, except the first and the last ones, are self-propelled by an active force with constant magnitude $F_a$: for monomer $i$, the force $\mathbf{F}_i^{\text{a}}$ reads

$$\mathbf{F}_i^{\text{a}} = F_a \frac{\mathbf{r}_{i+i} - \mathbf{r}_{i-i}}{|\mathbf{r}_{i+i} - \mathbf{r}_{i-i}|} \quad (3)$$

and it is parallel to the – normalized – backbone tangent[1, 41]. Activity is quantified via the adimensional Péclet number, which measures the strength of the self-propulsion in relation to the thermal noise, defined as

$$\text{Pe} = F_a \sigma / k_B T \quad (4)$$

Finally, the active polymers are confined in straight, cylindrical channels. Confinement is provided by as a collection of immobile beads of diameter $\sigma$ placed around the channel axis at a fixed distance $R + \sigma$ from it; the distance $R$ is, thus, the radius of the channel. The beads are arranged regularly and, at least, they are at contact distance with their nearest neighbours, as to prevent any possible escape of the polymer.

Throughout the work, we consider the monomers as having unitary mass $m$; we further set $\sigma$ and the thermal energy $k_BT$ as the units of length and energy, respectively, so that the characteristic simulation time $\tau$ is unitary. We perform Langevin Dynamics simulations, in the overdamped regime, disregarding hydrodynamics. We employ the open source package LAMMPS[68], with in-house modifications to implement the tangential activity. We integrate the equations of motion using the Velocity Verlet algorithm and choose time step $\Delta t = 10^{-3}\tau$. In order to ensure the overdamped regime within the range of values of Pe considered, following[42], we set the friction coefficient $\gamma$ to $\gamma = 1k_BT\tau/\sigma^2$, if Pe $< 1$ and $\gamma = 10k_BT\tau/\sigma^2$ if Pe $\geq 1$.
We simulate polymers consisting of $N$ monomers, with $40 < N < 750$, at different confinement conditions $6 \leq R/\sigma \leq 18$. We further vary the activity between $0.03 \leq \text{Pe} \leq 10$ and average over $M = 50 - 100$ independent realisations. The simulation box is orthogonal, with two short sides $L_y = L_z = 2(R + \sigma) + \sigma$ and a long side $L_x = N\sigma$, parallel to the channel axis, chosen to ensure a fully stretched polymer is contained in a single box. The simulation box is periodic along the channel ($x$) axis. Initial configurations are prepared from equilibrium simulations in good solvent conditions, performed using the same Kremer-Grest model. When activity is turned on,

we first perform simulation runs to reach the steady state, followed by production runs of (on average) $1 \cdot 10^6 \tau$, corresponding to $1 \cdot 10^9$ time steps (snapshots of the systems are taken every $\tau_s = 10^4 \tau$ or $10^7$ time steps).

### B. Structural and dynamical properties

We compute the average square end-to-end distance, defined as the square euclidean distance between the two ends of the polymer chain

$$\langle R_e^2 \rangle = \langle (\mathbf{r}_N - \mathbf{r}_1)^2 \rangle \tag{5}$$

where the average is performed, in steady state, on time and on different realisations. In the rest of the manuscript we will consider the (average) end-to-end distance $R_e = \sqrt{\langle R_e^2 \rangle}$.

We will also report the probability of finding the centre of mass of the polymer $\mathbf{r}_{\mathrm{cm}} = \frac{1}{N} \sum_{i=1}^N \mathbf{r}_i$ inside the channel at a distance $r$ from the centre of the channel, as well as the the orientation of the polymer, i.e. computing the angle between the axis of the channel (the $x$ axis in our simulations) and the instantaneous end-to-end vector, which reads

$$\theta = \arccos \left( \frac{\mathbf{R}_e \cdot \mathbf{x}}{|\mathbf{R}_e|} \right) \tag{6}$$

We remark that the probability distribution of the centers of mass is a radial distribution and, as such, we divide the measured probability by the area of the circular corona between $r$ and $r + \Delta r$, $\Delta r$ being the chosen bin width.

Concerning the dynamics of the chain, we compute the characteristic time scale of the dynamics, $\tau_e$, as the correlation time of the end-to-end vector: we extract $\tau_e$ from the end-to-end time autocorrelation function

$$C_{R_e}(t) = \left\langle \frac{\mathbf{R}_e(t_0 + t) \cdot \mathbf{R}_e(t_0)}{|\mathbf{R}_e(t_0)|^2} \right\rangle \tag{7}$$

by fitting the data at short times with an exponential function $C_{R_e}(t) = \exp(-t/\tau_e)$. The average (in steady state) in Eq. (7) is performed both on the initial time $t_0$ and on the ensemble of the different realisations. Finally, we compute the mean square displacement (MSD) of the centre of mass along the direction of the channel axis, i.e. along the $x$ axis

$$\langle \Delta x^2(t) \rangle = \left\langle [x_{\mathrm{cm}}(t_0 + t) - x_{\mathrm{cm}}(t_0)]^2 \right\rangle \tag{8}$$

where $x_{\mathrm{cm}}$ is the position of the centre of mass of the polymer along the channel axis. The average is performed as previously detailed. For times much longer than $\tau_a$, the MSD grows linearly in time. As common in active matter systems, we identify this regime as the long-time active diffusive regime and compute the (long time) active diffusion coefficient $D_a$ as

$$D_a = \lim_{t \to \infty} \frac{\langle \Delta x^2(t) \rangle}{2 d\, t} \tag{9}$$

following the Einstein relation, where $d = 1$ is the effective dimensionality of the system.

### C. Theoretical Modeling

#### 1. Mapping an active polymer to an Active Brownian Particle

The mapping between a tangentially-driven active polymer in bulk and an Active Brownian particle (ABP) has been introduced in Refs. [1, 42, 44]. We extend this mapping to describe the dynamics of an active polymer under confinement. Specifically, we map the dynamics of the centre of mass of the polymer, projected along the axis of the channel, to an ABP in 1D. Briefly, the ABP model is characterised by an active force, $\mathbf{f}_a$, whose magnitude $f_a$ is constant and a self-propulsion direction $\hat{n}$ evolving in time as a stochastic process with a characteristic time, $\tau_r$, usually called reorientational time. In what follows, we will consider an ABP model whose active force evolve by rotational diffusion. In addition, the ABP is subject to thermal noise, with thermal energy $k_B T$ and friction coefficient $\gamma$.

In the overdamped limit, the diffusion coefficient of an ABP can be expressed as[69]

$$D = D_t + \frac{\tau_r\, v_a^2}{2d} = D_t + \frac{\tau_r\, (f_a/\gamma)^2}{2d} \tag{10}$$

where $v_a = f_a/\gamma$ is the self-propulsion velocity and $D_t = k_B T/\gamma$ the translational diffusion coefficient. Since we consider an ABP in 1D, we set $d = 1$.

In order to map the polymer to an ABP, we have to provide effective values for the friction coefficient, the reorientation time and the active force. For a polymer of length $N$, the effective friction coefficient on the centre of mass is related to the monomer friction coefficient $\gamma_0$, disregarding hydrodynamics, as

$$\gamma_P = N\, \gamma_0 \tag{11}$$

For the polymer model considered in this study, we approximate the total active force $\mathbf{F}_a^P = \sum_i \mathbf{F}_i^a$ to be almost parallel to the end-to-end vector of the polymer, $\mathbf{R}_e$

$$\mathbf{F}_a^P \approx F_a \frac{\mathbf{R}_e}{\sigma} \tag{12}$$

where $\sigma$ and $F_a$ are defined in Sec. II A. As in Eq. (10), we define the propulsion speed for the effective ABP particle as the ratio between the magnitude of the total active force $\mathbf{F}_a^P$ and the total friction coefficient $\gamma_P$

$$v_a^P = \frac{|\mathbf{F}_a^P|}{\gamma_P} \approx \frac{F_a}{N\gamma_0} \frac{R_e}{\sigma} = \frac{1}{N\gamma_0} \frac{\mathrm{Pe}\, k_B T}{\sigma} \frac{R_e}{\sigma} \tag{13}$$

where we take $R_e = \sqrt{\langle R_e^2 \rangle} = |\mathbf{R}_e|$. The thermal contribution to the diffusion coefficient follows from the Rouse model,

$$D_t^P = \frac{k_B T}{\gamma_P} = \frac{k_B T}{N \gamma_0} = \frac{D_0}{N} \qquad (14)$$

$D_0 = k_B T / \gamma$ being the diffusion coefficient of single passive monomer. Finally, from Eq. (12) follows that the reorientational time of the active force is equal to the correlation time of the end-to-end vector of the polymer

$$\tau_r^P = \tau_e \qquad (15)$$

where the auto-correlation time of the end-to-end vector $\tau_e$ was introduced in Sec. II B.
Combining Eqs. (10),(12), (13) and (15), we can write

$$\frac{D_a - D_t^P}{D_t^P} = \frac{\tau_r^P D_t}{\sigma^2} \frac{R_e^2}{2N\sigma^2} \mathrm{Pe}^2 \qquad (16)$$

This expression is equivalent to that reported in [1, 42], but projected in 1D.

### 2. Predicting the diffusion coefficient: bulk estimate and angular correction under confinement

In the bulk, one can take advantage of the scaling relations, reported in [1]: the modulus of the end-to-end vector follows

$$R_e^B = \sigma \frac{a_{R_e} + h_{R_e} \ln\left(\sqrt{\delta_{R_e}^2 + \mathrm{Pe}^2}\right)}{(\mathrm{Pe} + 1)^{c_{R_e}}} N^{\nu(\mathrm{Pe})} \qquad (17)$$

In the Supplemental Material, we report measurements for $R_e$ in the bulk for the model discussed in Sec. II A; we verify that the functional form of Eq. (17) remains valid, within the range of parameter considered, with $a_{R_e} = 1.5$, $h_{R_e} = 0.058$, $\delta_{R_e} = 0.0005$, $c_{R_e} = 0.201$ and $\nu(\mathrm{Pe}) = 0.54 \, \mathrm{Pe}^{-0.022}$. Further, an expression for the correlation time of the end-to-end vector is available in the bulk:

$$\frac{\tau_e D_0}{\sigma^2} = \tau_0^B \frac{N}{\mathrm{Pe}} \qquad (18)$$

with $\tau_0^B = 0.5$. Thus, if we disregard the effect of confinement, we get the following expression:

$$\frac{D_a^B - D_t^P}{D_t^P} = \frac{\tau_0^B}{2} \frac{\mathrm{Pe}}{(\mathrm{Pe} + 1)^{2c_{R_e}}} \cdot$$
$$\cdot \left( a_{R_e} + h_{R_e} \ln\left(\sqrt{\delta_{R_e}^2 + \mathrm{Pe}^2}\right) \right)^2 N^{2\nu(\mathrm{Pe})} \qquad (19)$$

Finally, we include the contribution of confinement by adapting the correction developed in [70] for rod-shaped particles. We indeed approximate the polymer as a rod whose main axis is given by the end-to-end vector $\mathbf{R}_e$; the maximum angle between $\mathbf{R}_e$ and the channel axis

indicates the degree of confinement and is used to correct the estimation of the diffusion coefficient. The expression for the diffusion coefficient now reads:

$$\frac{D_a^C - D_t^P}{D_t^P} \approx (1 + 2\cos(\theta_{\max})) \; \pi \frac{\tau_e^C D_0}{\sigma^2} \frac{\left(R_e^C\right)^2}{3 N \sigma^2} \; \mathrm{Pe}^2 \qquad (20)$$

where $R_e^C$ is the end-to-end distance under confinement and $\sin(\theta_{max}) = h_0/R_e^C = 2R/R_e^C$. We highlight the addition, with respect to Eq. (16), of a factor $2\pi$ to include the contribution caused by the axial symmetry. We remark that $R_e^C$ and $\tau_e^C$ are estimated from simulation data (see Sec. II B). We report a detailed derivation of Eq. (20) in the Supplementary Material.

## III. RESULTS

We first report the scaling properties of the end-to-end vector and discuss a deviation of the numerical results with respect to the passive universal scaling under confinement. We further investigate the position of the polymer inside the channel and its orientation with respect to the channel axis. Then we report on the dynamics of the active filaments, discussing the scaling of the correlation times. Finally, we apply the theoretical mapping, introduced in Sec. II C and compare the predictions against the numerical data. For most observables, we report additional data in the Supplemental Material.

### A. Scaling properties of the end-to-end distance

First, we report on the scaling properties of the end-to-end distance $R_e$ for active polymers under confinement. $R_e$ provides a measure of the polymer extension, taken to be representative of the polymer size under cylindrical confinement. Indeed, passive polymer filaments under uniaxial confinement become more aspherical, as the channel restricts the available space in the plane transverse to its axis. We thus focus on characterising its scaling properties, i.e. how $R_e$ grows upon increasing the number of monomers $N$. For confinement lengths smaller than (or comparable to) the extension of the polymer[65], the scaling of $R_e$ for passive, flexible polymers under confinement is characterised by the de Gennes regime.

For tangentially active polymers, the scaling of $R_e$ in the bulk does not follow the same power-law as in passive systems: a coil-to-globule-like transition appears upon increasing the activity[1, 41, 44, 46]. This implies a decrease of the value of $\nu$, as compared to the passive value. We should thus expect an interplay between the two opposite trends imposed by activity and confinement. We first discuss, in Fig. 2, the dependence of $R_e$ on $N$ for tangentially active linear chains under confinement in two limiting cases of weak ($\mathrm{Pe} \ll 1$) and strong ($\mathrm{Pe} \gg 1$) activity. Specifically, we consider $\mathrm{Pe} = 0.03$ ( Fig. 2A) and $\mathrm{Pe} = 10$ (Fig. 2B); we also report

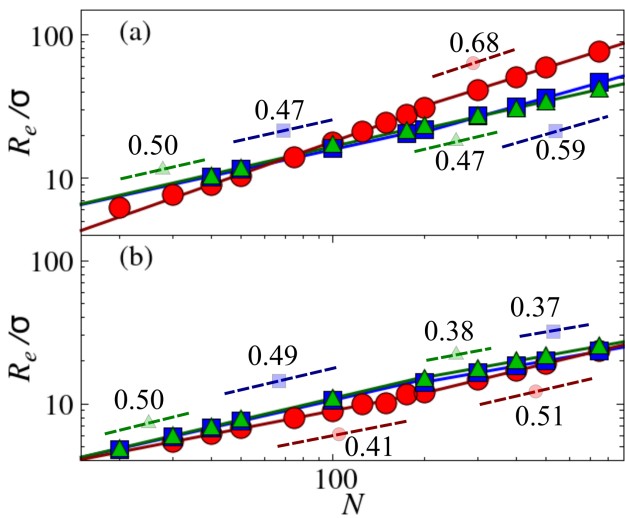

FIG. 2. End-to-end distance $R_e/\sigma$ as a function of $N$ at fixed Pe =0.03 (panel a) and Pe = 10 (panel b) for different values of $R = 6$ (red circles), 13 (blue squares) and 18 (green triangles). The slopes shown in each panel have been computed by fitting each set of points independently for $N < \tilde{N} = 200$ and $N > \tilde{N} = 200$.

the measured exponents for all the values of $R$ and Pe in the Supplemental Material. We first examine the strongest confinement considered ($R = 6\sigma$, red symbols in Fig. 2). In the limit of weakly active chains (panel a), a single power law emerges; the exponent $\nu$ depends on the value of Pe and it is found to be always larger than the (passive) bulk exponent. Indeed, here $\nu$ is reminiscent of the passive exponent under confinement ($\nu = 1$); however, as the tangential activity tends to shrink the polymer chains, the final outcome results from their interplay.

Upon increasing activity (panel b), we observe the emergence of two distinct power law regimes, with exponents $\nu_1$ and $\nu_2$. At high activity polymers become more compact and, as such, confinement becomes less severe, for the same degree of polymerisation $N$. In other words, the same channel is effectively larger for polymers at higher activity. However, we find that the first exponent, valid for chains below $\tilde{N} \simeq 200$, is $\nu_1 \simeq 0.40$, smaller than the bulk exponent $\nu_b \simeq 0.51$ at the same value of Pe. Confinement thus seems to enhance compaction for short, highly active polymers. We may speculate that, under tight confinement conditions, short active chains fold and collapse even more than in bulk due to the presence of hard walls with a rather pronounced curvature. On the other hand, above $\tilde{N}$, the exponent $\nu_2$ is found to be larger than $\nu_1$, having a value that is compatible with $\nu_b$.

Upon increasing the value of the channel radius $R$ we observe that two power-law regimes emerge also in the weak activity limit (Fig. 2a, squares and triangles). Again, the phenomenology observed depends from the value of $R$ and Pe but, as expected, the effect of activity becomes, upon releasing the confinement, more dominant for shorter polymers at any value of Pe. However, the exponent $\nu_1$ remains smaller than its bulk counterpart $\nu_b(\text{Pe})$ for Pe $\ll 1$. Only at large values of $R$ and Pe (Fig. 2b) we recover $\nu_1 = \nu_b(\text{Pe})$. It is interesting to remark that, in a passive system, a bulk-like regime would already be detectable, even at $R = 13\sigma$: the absence of such regime is a signature of the non-trivial interplay between activity and confinement.

Focusing now on longer polymers, we further detect, for $R = 13\sigma$ and Pe $\ll 1$, that $\nu_2 > \nu_1$, implying that the polymer size is more sensitive to the length of the backbone. Instead for Pe $\gg 1$, at the same confinement, $\nu_2 < \nu_1$, i.e. the polymer size is less sensitive to the length of the backbone.
We can again ascribe these different behaviours to the interplay between activity and confinement: the effects of confinement become dominant over the effects of activity for Pe $\ll 1$, vice-versa for Pe $\gg 1$, at least within a non-negligible window of polymer sizes. Notably, for Pe $\gg 1$, confinement seems to enhance the globule-like character of the tangential propulsion. We

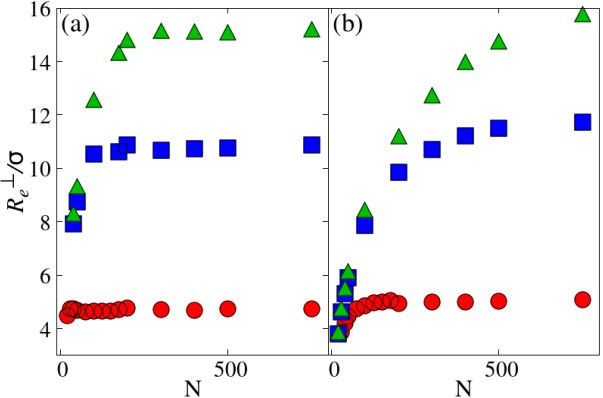

FIG. 3. $R_e^{\perp}$ as a function of $N$ for active polymer chains for $R = 6$ (red circles), 13 (blue squares) and 18 (green triangles), and (a) Pe = 0.03, (b) Pe = 10.

get more insight by studying $R_e^{\perp}$, i.e. the component of $R_e$ perpendicular to the channel axis, as reported in Fig. 3 as a function of $N$ for various values of $R$. We observe that, at low activity, $R_e^{\perp}$ reaches a saturation value roughly at the same value of $N$ ($\tilde{N} \approx 200$), except under very strong confinement where $R_e^{\perp}$ is constant for all values of $N$ considered. At high activity, the saturation value is reached at larger values of $N$: for example, at the tightest confinement, we observe a saturation at $\tilde{N} \approx 200$, curiously the same value found

at small activities. We can thus better interpret the transitions to different regimes in Fig. 2, at least in the low Pe limit: as the transversal size of the polymers becomes as large as the channel width, active chains can only increase in the direction parallel to the channel axis. Conversely, if the transverse size is, within the range of values of $N$ considered, always at saturation, than only one regime will be observed. This correspondence holds true also for Pe $\gg 1$ and tight confinement conditions. Upon increasing $R$ at high Pe, $R_e^\perp$ suggests that the confinement-enhanced compaction observed at tight confinement will shift to even higher values of $N$ than the ones considered in this work.

It is also interesting to recast the same data in a different fashion. Indeed, passive flexible polymers display a universal scaling of $R_e$ under confinement[66, 71]: in particular, the ratio of the magnitude of the end-to-end vector under confinement over its bulk counterpart $R_e/R_e^b$ is a universal function of $R/R_e^b$. It is thus

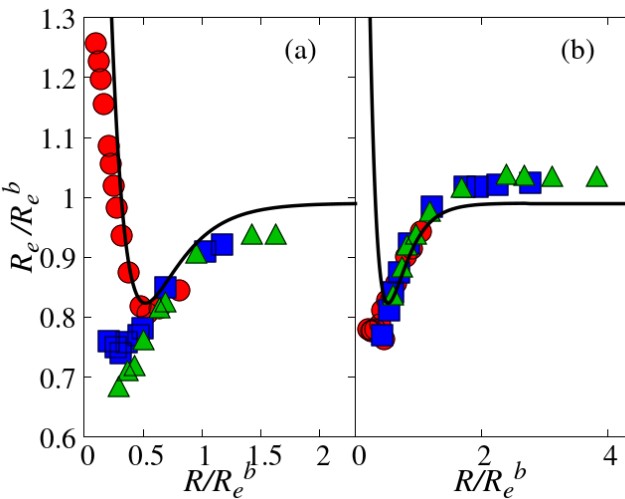

FIG. 4. End-to-end distance $R_e$, scaled over the bulk value $R_e^b$ at the same Pe as a function of $R/R_e^b$ at fixed values of Pe $=0.03$ (panel a) and Pe $=10$ (panel b), for different confinement conditions, $R=6$ (red circles), 13 (blue squares) and 18 (green triangles).

compelling to assess if such scaling holds for active polymers, where $R_e^b = R_e^b(N, \text{Pe})$ is given by Eq. (17) in the active case. We report the result in Fig. 4, where we plot $R_e/R_e^b$ as a function of $R/R_e^b$ for Pe $= 0.03, 10$ and different values of $R$. We observe that, in some regimes of confinement and activity, sufficiently small active polymers behave as their passive counterpart (gray line). In contrast, at low confinement or at very high activity, the rescaled data do not follow the passive master curve. Further, long enough polymers deviate from the scaling for any value of Pe or $R$. More importantly, given this rescaling, the reported data do not collapse on a single universal curve.

The universal behaviour of passive flexible polymers under confinement follows the de Gennes blob picture. The characteristic length scale of the blob is either the thermal one or it is set by the confinement; the interplay between the two determines the master curve[65, 66, 71]. As reported for another active polymer model under confinement[60], a straightforward conclusion is the failure of a blob description, i.e. it is not possible to uniquely define a correlation blob, as other relevant length scales arise. This further hints at the fact that the polymer self-similarity is broken at some non-trivial scale; while the model is different, the effect reported here is similar to what was observed for Active Brownian Polymers[60].

## B. Radial and angular distributions

We further characterize the polymer conformations by looking at the position and orientations of the chains inside the channel. Specifically, we look at the distribution of the centre of mass within the channel and at the distribution of the angle between the end-to-end vector and the channel axis (see Sec. II B).

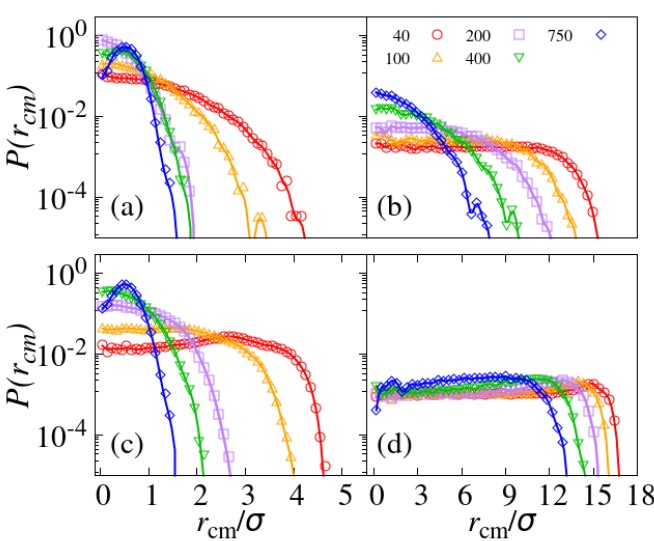

FIG. 5. Distribution of the position of the centre of mass of the polymers as a function of the radial distance from the channel centre for different values of N and (a): $R = 6\sigma$ Pe $= 0.03$ (b): $R = 18\sigma$ Pe $= 0.03$ (c): $R = 6\sigma$ Pe $= 10$ and (d): $R = 18\sigma$, Pe $= 10$.

Concerning the former observable, we focus on four specific cases. In Fig. 5 we report the distribution of the centre of mass positions as a function of the distance from the channel axis, measured in steady state, upon varying $N$, Pe, and $R$. In all plots, the channel walls are at $r = R$. We observe that, under strong confinement conditions and weak activity (panel a), the centre of mass of the polymers does not accumulate at the

boundary. Large polymers occupy the whole channel, as already pointed out, and the centre of mass is located near the centre of the channel. However, no excess probability at the boundary is found for small polymers. Looking at the probability distribution of the individual monomers, that we report in the Supplemental Material, one still does not find strong evidence for wall accumulation. Upon increasing $R$ or Pe (panels b-d) the wall accumulation partially reappears. In general, the fact that, upon increasing $N$, polymers grow in the transverse direction (see Fig. 3), until their transversal size reaches the channel width, forces the centre of mass to be located in the middle of the channel even for Pe $\gg$ 1 and in mild confinement. This effect makes flexible active filaments stand out in comparison with active colloids. Indeed, active polymers do not accumulate at the channel boundary as long as the size of the channel $R$ is not much larger than the polymer size, in which case the filament can be reasonably approximated by an effective soft colloid.

Further, as anticipated in Sec. II C, in order to ratio-

the distribution is rather flat; this is expected, as the polymer is not constrained, in this instance, to assume any particular orientation by the confinement. On the contrary, large polymers do preferentially align with the channel axis ($\theta = 0, \pi$); the effect is relevant under strong confinement or weak activity conditions. We can thus conclude that the orientations of the polymer are limited only for large values of $N$ and strong confinement conditions, i.e when $R_e(N) \simeq R$. The condition is indeed captured by the definition of $\theta_{max}$ given in Sec. II C.

### C. Scaling of the correlation time

We further report the correlation time of the chains; this is, as in [1, 42, 44], the characteristic time of the self-propulsion force and the characteristic time of the active contribution to the diffusion coefficient (see Sec. II C).

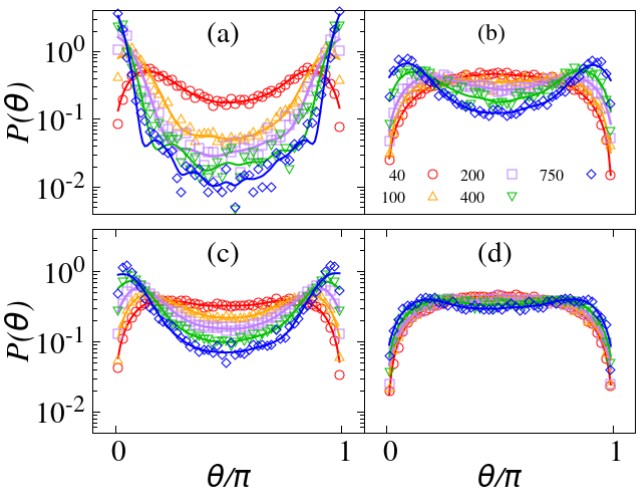

FIG. 6. Distribution of the angles between the instantaneous end-to-end vector and the channel axis for different values of $N$ and (a): $R = 6\sigma$ Pe $= 0.03$ (b): $R = 6\sigma$ Pe $= 10$ (c): $R = 18\sigma$, Pe $= 0.03$ (d) $R = 18\sigma$, Pe $= 10$

nalise the transport properties of tangentially propelled polymers under confinement, we will approximate the polymer as a rigid rod and assume that its orientations is limited. We double check that such approximation is meaningful, assessing how the polymers are oriented with respect to the channel axis (see Sec. II B). In Fig. 6, we report the distribution of the angles between the end-to-end vector and the axis of the channel, measured in steady state; the panels (a)-(d) refer to the same cases discussed above. We remark that, if $\theta = 0$ or $\theta = \pi$, the end-to-end vector lies parallel to the channel axis; conversely, if $\theta = \pi/2$, the the end-to-end vector is perpendicular to it. We observe that, in all four cases, if the polymer size is small with respect to $R$,

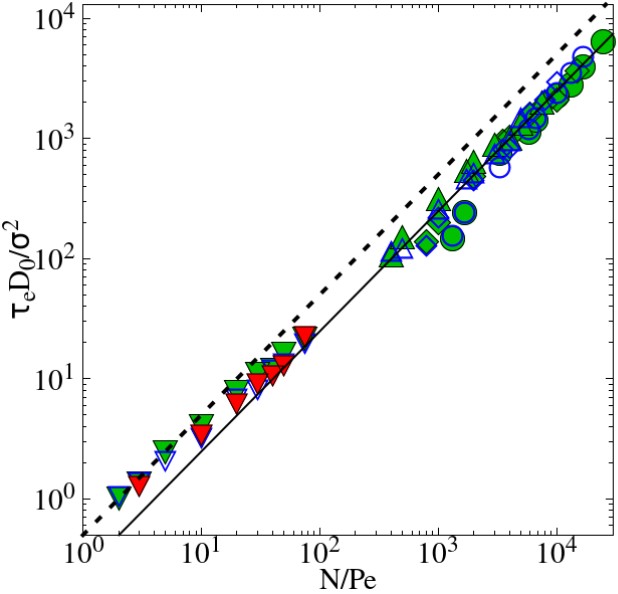

FIG. 7. End-to-end vector correlation time $\tau_e D_0/\sigma^2$ as a function of $N$/Pe for different values of Pe $=0.03$ (circle), 0.05 (diamont), 0.1 (triangle), 10 (inverted triangle) and confinement $R =6$ (red), 13 (blue), 18 (green). Dashed line represents the prediction from the Bulk ($\tau_0 = 0.5$) and the continuous line represents the linear fit to these data ($\tau_0 = 0.25$).

In Fig. 7 we report the adimensional correlation time $\tau_e D_0/\sigma^2$ as a function of $N$/Pe, in the same fashion of [1]. We observe that a linear dependence is still valid. However, with respect to the bulk scaling (dashed line), data under confinement have the same scaling but a different slope. Indeed, given

$$\tau_e D_0/\sigma^2 = \tau_0 \frac{N}{\text{Pe}} \qquad (21)$$

we obtain $\tau_0 \simeq 0.25$ under confinement, smaller than the bulk value $\tau_0^B \simeq 0.5$. We can recast Eq. (21) in terms of the self-propulsion velocity of the polymer, $v_a = f_a/\gamma$, as

$$\tau_e = \tau_0 \frac{L}{v_a} \qquad (22)$$

This relation shows that, even under confinement, the polymer is still driven by a "railway motion", as named in the literature[38, 44]. The influence of the confinement on the correlation time amounts, thus, to a decrease of the decorrelation time.

### D. Diffusion coefficient predictions via the ABP mapping

In this section, we present results on the transport properties, namely the diffusion coefficient along the channel axis, of active polymers under cylindrical confinement. We will compare the results of the numerical simulations with the predictions of the two models proposed in Sec. II C Eq. (19) and Eq. (20).

We report the comparison in Fig. 8. Again, we focus on two extreme cases of strong confinement $R = 6\sigma$ (panel a) and weak confinement (panel b). The comparison with the theoretical approximations Eq. (19) shows that, under strong confinement, the diffusion coefficient can be enhanced by roughly a factor of 10. The geometrical correction, introduced in Sec. II C, nicely agrees with the simulation data. On the other hand, when the confinement is weak, the bulk prediction remains in very good agreement with the numerical data, see Fig.8b. We notice that the discrepancy with the bulk prediction also hints at the fact that the diffusion now increases as a function of $N$. Further, we may also observe that some of the characteristic features of these active polymers are maintained in confinement as well. Indeed, we report the data in the same fashion as in [1] in the Supplemental Material. One can appreciate that $D_a^C/D_0$ ($D_0$ being the diffusion coefficient of a single monomer) increases upon increasing Pe and, for large values of Pe and weak confinement conditions, the diffusion coefficient is roughly independent on $N$. However, as already mentioned, the diffusion coefficient increases with $N$ under strong confinement.

### IV. CONCLUSIONS

We report on the conformation and dynamics of tangentially active polymers under cylindrical confinement. Concerning the former, on the one hand one expects strong confinement to induce elongated conformations. On the other hand, strong self-propulsive forces tend to drive the polymer collapse, as observed in bulk. It is important to stress that the same channel can stand as a mild or strong confinement, depending on the polymer's degree of polymerisation or, in general, on its

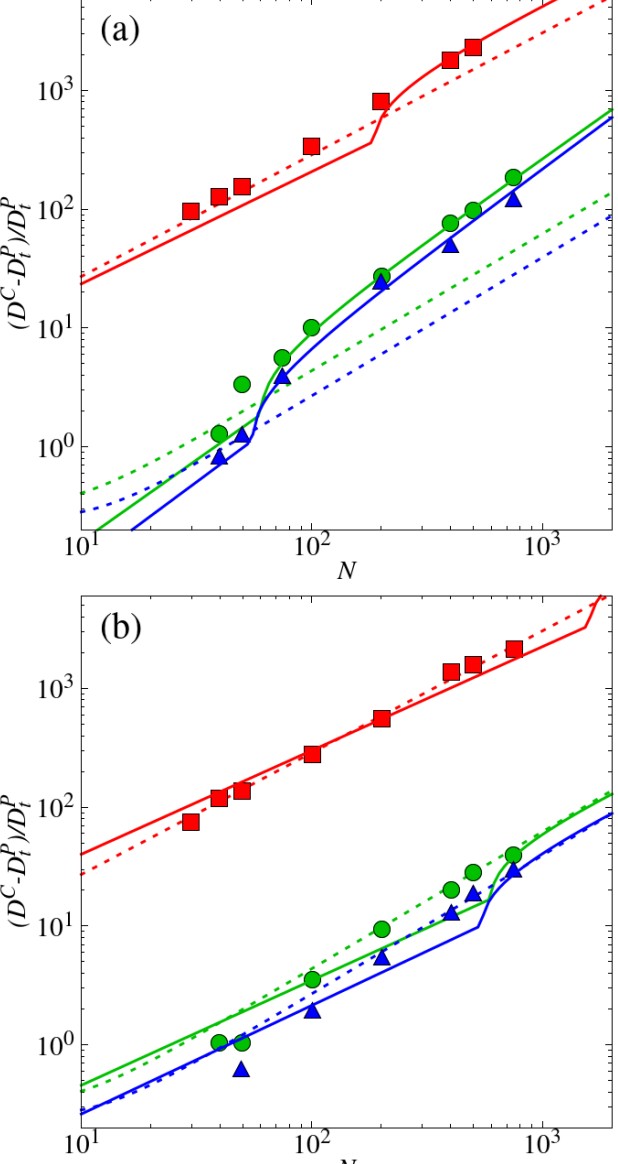

FIG. 8. Long time diffusion coefficient as a function of $N$ for different values of Pe = 0.03 (blue triangles), 0.05 (green circles), 10 (red squares) and (a) $R = 6\sigma$ (b) $R = 18\sigma$. The symbols refer to simulation data, lines are the theoretical predictions, dashed lines referring to Eq. (19), full lines to Eq. (20).

size. Indeed, we observe a complex interplay between activity and confinement, with non trivial power law regimes emerging at different values of the activity and at different confinement conditions. Further, by checking that the no universal curve can be obtained upon rescaling, we conclude that the blob picture is broken for active polymer and that the channel radius cannot be taken as the fundamental length scale of the system.

We further show that the tendency of active polymers

to accumulate at the boundary of the channel is qualitatively different from the colloidal case, which may be a feature that makes soft, deformable active filaments stand apart from active colloids. It is intriguing to notice that the accumulation has been observed for many elongated swimmers, such as sperms. As such, it would be interesting from a biophysical perspective to assess more in detail what are the minimal requirements for wall aggregation, in terms of aspect ratio and filament flexibility, having focused here to relatively large aspect rations ($N > 40$). Wall aggregation is indeed argued to be important for early stage biofilm formation[72].

Finally, from the perspective of the dynamics, we show that there is a significant deviation from the bulk, regarding the correlation times and the diffusion coefficient. For the latter, we propose a correction, based on a geometrical argument. The correction is, in principle, valid for rigid rods, which is not the case for the polymers under investigation. However, as the total active force is almost parallel to $R_e$, approximating the polymer as its end-to-end vector well captures the physics of the system. Indeed, such a correction yields a good comparison against the simulation results, without any fitting parameter.

The study of active polymers under confinement may be relevant in different contexts. Many filamentous organisms live under strong confinement conditions; understanding how they behave may be important to create bio-mimetic soft robots able to burrow and perform tasks, as worms do[73]. Further, porous media are used to filter and separate passive polymers; possibly a similar principle could be devised for active filaments such as cyanobacteria, possibly using channels with varying section[74] or with specific asymmetries.

## ACKNOWLEDGMENTS

E. Locatelli acknowledges support from the MIUR grant Rita Levi Montalcini and from the HPC-Europa3 program. C.V. acknowledges fundings IHRC22/00002 and PID2022-140407NB-C21 from MINECO. The computational results presented have been achieved using the Vienna Scientific Cluster (VSC) and the Barcelona Supercomputing Center (BSC-Marenostrum); CloudVeneto is also acknowledged for the use of computing and storage facilities.

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

[1] V. Bianco, E. Locatelli, and P. Malgaretti, Globulelike conformation and enhanced diffusion of active polymers, Physical Review Letters **121**, 217802 (2018).

[40] S. K. Anand and S. P. Singh, Structure and dynamics of a self-propelled semiflexible filament, Phys. Rev. E **98**, 042501 (2018).

[41] M. Foglino, E. Locatelli, C. Brackley, D. Michieletto, C. Likos, and D. Marenduzzo, Non-equilibrium effects of molecular motors on polymers, Soft matter **15**, 5995 (2019).

[42] M. Fazelzadeh, E. Irani, Z. Mokhtari, and S. Jabbari-Farouji, Effects of inertia on conformation and dynamics of tangentially driven active filaments, Phys. Rev. E **108**, 024606 (2023).

[43] C. A. Philipps, G. Gompper, and R. G. Winkler, Tangentially driven active polar linear polymers—an analytical study, The Journal of Chemical Physics **157** (2022).

[44] J.-X. Li, S. Wu, L.-L. Hao, Q.-L. Lei, and Y.-Q. Ma, Nonequilibrium structural and dynamic behaviors of polar active polymer controlled by head activity, Physical Review Research **5**, 043064 (2023).

[45] D. Breoni, C. Kurzthaler, B. Liebchen, H. Löwen, and S. Mandal, Giant activity-induced stress plateau in entangled polymer solutions, arXiv preprint arXiv:2310.02929 (2023).

[46] P. Malgaretti, E. Locatelli, and C. Valeriani, Coil-to-globule collapse of active polymers: a rouse perspective, arXiv preprint arXiv:2404.12470 (2024).

[47] M. A. Ubertini, E. Locatelli, and A. Rosa, Universal time and length scales of polar active polymer melts, arXiv preprint arXiv:2404.08425 (2024).

[48] T. Terakawa, S. Bisht, J. M. Eeftens, C. Dekker, C. H. Haering, and E. C. Greene, The condensin complex is a mechanochemical motor that translocates along dna, Science **358**, 672 (2017).

[49] R. K. Manna and P. S. Kumar, Emergent topological phenomena in active polymeric fluids, Soft Matter **15**, 477 (2019).

[50] S. Das and A. Cacciuto, Dynamics of an active semiflexible filament in a spherical cavity, The Journal of Chemical Physics **151** (2019).

[51] I. Chubak, S. M. Pachong, K. Kremer, C. N. Likos, and J. Smrek, Active topological glass confined within a spherical cavity, Macromolecules **55**, 956 (2022).

[52] A. Mahajan, W. Yan, A. Zidovska, D. Saintillan, and M. J. Shelley, Euchromatin activity enhances segregation and compaction of heterochromatin in the cell nucleus, Physical Review X **12**, 041033 (2022).

[53] Z. Mokhtari and A. Zippelius, Dynamics of active filaments in porous media, Physical review letters **123**, 028001 (2019).

[54] C. Kurzthaler, S. Mandal, T. Bhattacharjee, H. Löwen, S. S. Datta, and H. A. Stone, A geometric criterion for the optimal spreading of active polymers in porous media, Nature communications **12**, 7088 (2021).

[55] L. Theeyancheri, S. Chaki, T. Bhattacharjee, and R. Chakrabarti, Migration of active rings in porous media, Physical Review E **106**, 014504 (2022).

[56] Y. Wang, Y.-w. Gao, K. Chen, *et al.*, Obstacle-induced giant jammed aggregation of active semiflexible filaments, Physical Chemistry Chemical Physics **24**, 23779 (2022).

[57] M. Fazelzadeh, Q. Di, E. Irani, Z. Mokhtari, and S. Jabbari-Farouji, Active motion of tangentially driven polymers in periodic array of obstacles, The Journal of Chemical Physics **159**, 224903 (2023).

[58] F. Tan, R. Yan, C. Zhao, and N. Zhao, Translocation dynamics of an active filament through a long-length scale channel, The Journal of Physical Chemistry B **127**, 8603 (2023).

[59] J. P. Miranda, E. Locatelli, and C. Valeriani, Self-organized states from solutions of active ring polymers in bulk and under confinement, Journal of Chemical Theory and Computation **20**, 1636 (2024).

[60] S. Das and A. Cacciuto, Deviations from Blob Scaling Theory for Active Brownian Filaments Confined Within Cavities, Physical Review Letters **123**, 087802 (2019).

[61] C. M. Barriuso Gutiérrez, J. Martín-Roca, V. Bianco, I. Pagonabarraga, and C. Valeriani, Simulating microswimmers under confinement with dissipative particle (hydro) dynamics, Frontiers in Physics **10**, 926609 (2022).

[62] C. Li, Q. Chen, and M. Ding, Escape dynamics of active ring polymers in a cylindrical nanochannel, Soft Matter (2024).

[63] D. Marenduzzo, C. Micheletti, and E. Orlandini, Biopolymer organization upon confinement, Journal of Physics: Condensed Matter **22**, 283102 (2010).

[64] M. Muthukumar, Polymers under confinement, Advances in Chemical Physics **149**, 129 (2012).

[65] L. Dai, C. B. Renner, and P. S. Doyle, The polymer physics of single dna confined in nanochannels, Advances in colloid and interface science **232**, 80 (2016).

[66] G. Morrison and D. Thirumalai, The shape of a flexible polymer in a cylindrical pore, The Journal of chemical physics **122**, 194907 (2005).

[67] K. Kremer and G. S. Grest, Dynamics of entangled linear polymer melts: A molecular-dynamics simulation, The Journal of Chemical Physics **92**, 5057 (1990).

[68] S. Plimpton, Fast parallel algorithms for short-range molecular dynamics, Journal of Computational Physics **117**, 1 (1995).

[69] B. ten Hagen, S. van Teeffelen, and H. Löwen, Brownian motion of a self-propelled particle, Journal of Physics: Condensed Matter **23**, 194119 (2011).

[70] P. Malgaretti and J. Harting, Transport of neutral and charged nanorods across varying-section channels, Soft Matter **17**, 2062 (2021).

[71] Y.-L. Chen, M. Graham, J. De Pablo, G. Randall, M. Gupta, and P. Doyle, Conformation and dynamics of single dna molecules in parallel-plate slit microchannels, Physical Review E **70**, 060901 (2004).

[72] J. Jara, F. Alarcón, A. K. Monnappa, J. I. Santos, V. Bianco, P. Nie, M. P. Ciamarra, Á. Canales, L. Dinis, I. López-Montero, *et al.*, Self-adaptation of pseudomonas fluorescens biofilms to hydrodynamic stress, Frontiers in Microbiology **11**, 588884 (2021).

[73] A. Kudrolli and B. Ramirez, Burrowing dynamics of aquatic worms in soft sediments, Proceedings of the National Academy of Sciences **116**, 25569 (2019).

[74] E. Locatelli, V. Bianco, C. Valeriani, and P. Malgaretti, Nonmonotonous translocation time of polymers across pores, Physical Review Letters **131**, 048101 (2023).

# Tangentially Active Polymers in Cylindrical Channels: Supplemental Material

José Martín-Roca, Emanuele Locatelli, Valentino Bianco, Paolo Malgaretti, Chantal Valeriani

## S1. ANGULAR CORRECTION TO THE DIFFUSION COEFFICIENT

We start from the overdamped Langevin equation of motion, projected on the longitudinal direction of the channel (x-axis)

$$\frac{d\vec{r}}{dt} \cdot \hat{x} = v_p \, \hat{n} \cdot \hat{x} + \sqrt{2D_t} \, \vec{\xi} \cdot \hat{x} \tag{S1}$$

We integrate Eq. (S1) and we obtain

$$\Delta\vec{r} \cdot \hat{x} = v_p \int_0^t \hat{n}(t') \cdot \hat{x} \, dt' + \sqrt{2D_t} \int_0^t \vec{\xi}(t') \cdot \hat{x} \, dt' \tag{S2}$$

then, forgetting cross-correlation terms and averaging over the ensemble, we get

$$(\Delta\vec{r} \cdot \hat{x})^2 = v_p^2 \int_0^t \int_0^t [\hat{n}(t') \cdot \hat{x}] \, [\hat{n}(t'') \cdot \hat{x}] \, dt' \, dt'' + 2D_t \int_0^t \int_0^t \left[\vec{\xi}(t') \cdot \hat{x}\right] \left[\vec{\xi}(t'') \cdot \hat{x}\right] dt' \, dt'' \tag{S3}$$

$$\langle(\Delta\vec{r} \cdot \hat{x})^2\rangle = v_p^2 \int_0^t \int_0^t \langle\cos[\theta(t')] \cdot \cos[\theta(t'')]\rangle \, dt' \, dt'' + 2D_t \int_0^t \int_0^t \langle\xi_x(t') \, \xi_x(t'')\rangle \, dt' \, dt'' \tag{S4}$$

The latter term yields the usual contribution $2D_t$. We further approximate the correlation term $\langle\cos[\theta(t')] \cdot \cos[\theta(t'')]\rangle$ as

$$\langle\cos[\theta(t')] \cdot \cos[\theta(t'')]\rangle \simeq \frac{1 + 2 \cos\theta_{max}}{3} \exp\left[-\frac{|t' - t''|}{\tau_r}\right], \langle\xi_x(t') \, \xi_x(t'')\rangle = \delta(t' - t'') \tag{S5}$$

where $\sin\theta_{max} = h_0/|R_e|$. Putting the result Eq. (S5) into Eq. (S4) we obtain

$$\langle(\Delta\vec{r} \cdot \hat{x})^2\rangle = \left[\frac{1 + 2 \cos\theta_{max}}{3} \frac{v_p^2}{D_r} + 2 \, D_t\right] t = 2 \, D_{eff} \, t \tag{S6}$$

We further correct Eq. (S6) by adding a factor $2\pi$, in order to include the contribution caused by the axial symmetry. We thus obtain

$$D_{eff} = D_t + [1 + 2 \cos(\theta_{max})] \, \frac{\pi}{3} \, \tau_r \, v_p^2 \tag{S7}$$

We remark that when the channel radius, $h_0$ is much larger than the end-to-end distance, $|R_e|$ we have $\theta_{max} = \pi/2$ and the angular correction in the last expression reads

$$\frac{1 + 2 \cos\theta_{max}}{3} = \frac{1}{3} \tag{S8}$$

i.e., the diffusion along a specific axis is $1/3$ of the overall diffusion. At variance, when $R_e \gg h_0$ we have $\theta_{max} \to 0$ and hence the angular correction reads

$$\frac{1 + 2 \cos\theta_{max}}{3} = 1 \tag{S9}$$

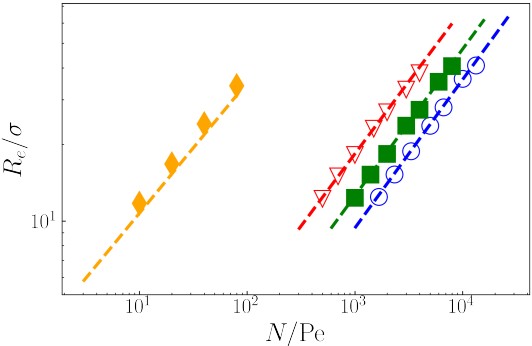

FIG. S1. Bulk end-to-end distance as a function of $N/\mathrm{Pe}$ for different values of Pe.

## S2. SCALING OF THE END-TO-END DISTANCE

In this section, we report further information concerning the scaling of the end-to-end vector. In particular, we report, in Fig. S1, the scaling of $R_e$ in the bulk. Indeed, as happens for passive polymers, the scaling pre-factors are not universal and should be computed upon changing the model. We show, in Fig. S1, that a functional form for $R_e$ of the same kind as the one adopted in [1], with slightly different coefficients, reproduced the numerical data very well, at least for the value of Pe considered.

Next, we report complementary data for $R_e$ as a function of $N$ under confinement in Fig. S2. In particular, we report

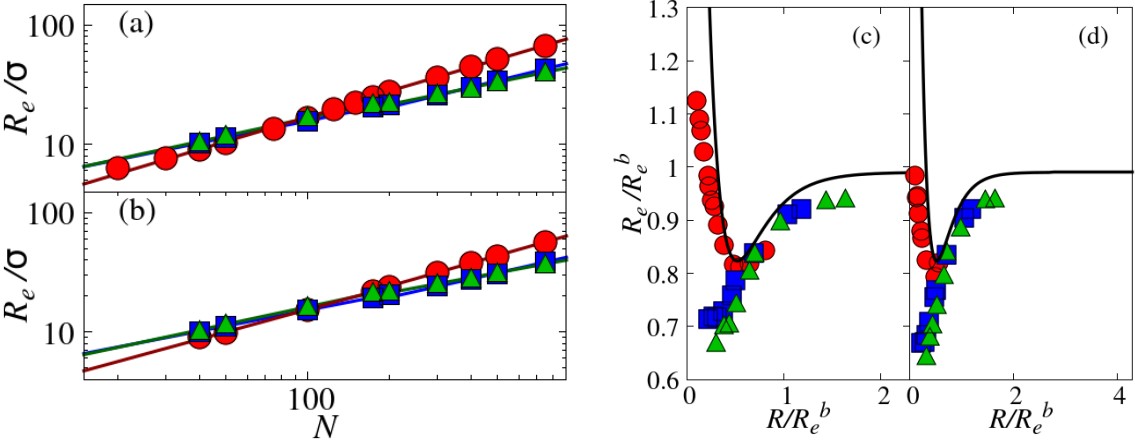

FIG. S2. (a)-(b): End-to-end distance $R_e/\sigma$ as a function of $N$ at fixed (a) Pe = 0.05 and (b) Pe = 0.1 for different values of $R$. (c)-(d): End-to-end distance $R_e$, scaled over the bulk value $R_e^b$ at the same Pe as a function of $R/R_e^b$ at fixed values of (c) Pe =0.05 and (d) Pe =0.1, for different confinement conditions.

the data for different values of $R$ at fixed Pe = 0.05 (panel a) and Pe = 0.1 (panel b). We can observe the same behaviour reported for Pe = 0.03 in the main text. Further, we report $R_e/R_e^b$ as function of $R/R_e^b$ for the same data in Fig. S2c,d. We observe also here that the data do not follow the universal scaling curve, valid for passive polymers; further, they do not collapse on a single master curve, as also reported in the main text for different values of Pe.

We report, in Fig. S3, $R_e^\perp$, i.e. the component of $R_e$ perpendicular to the channel axis, as a function of $N$ for different values of Pe and $R$. We observe that, at low activity, $R_e^\perp$ reaches a saturation value roughly at the same value of $N$ ($\tilde{N} \approx 200$), except under very strong confinement where $R_e^\perp$ is constant for all the values of $N$ considered. Such a saturation value is typically smaller than the channel radius $R$ ($R_e^\perp \simeq 0.85R$) if $Pe \ll 1$, while it becomes more similar to $R$ at high activity. At high activity, saturation also is reached at larger values of $N$: even at the tightest confinement, we observe a saturation at $\tilde{N} \approx 200$. As mentioned in the main text, this is connected to the globule-like collapse, driven by the tangential forces, that becomes relevant at high activity. We further report, in Table S1, all the different exponents $\nu_1$ and $\nu_2$, measured at different activity and confinement conditions. As reported, we notice a correspondence between the saturation of $R_e^\perp$ and the measured exponents. For example, at low activity and tight

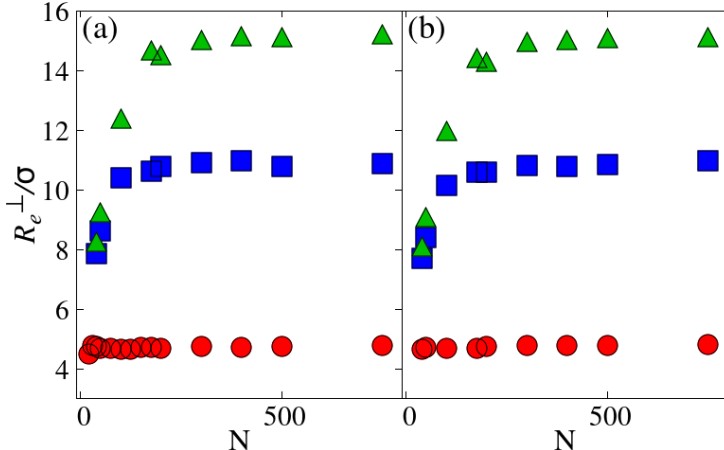

FIG. S3. $R_e^\perp$ as a function of $N$ for active polymer chains for different values of $R$ and (a) Pe $= 0.05$, (b) Pe $= 0.1$

| Pe | R | $\nu_1$ | $\nu_2$ | R | $\nu_1$ | $\nu_2$ | R | $\nu_1$ | $\nu_2$ |
|---|---|---|---|---|---|---|---|---|---|
| 0.03 | 6 | 0.7056 | 0.6753 | 13 | 0.4716 | 0.5916 | 18 | 0.4983 | 0.4724 |
| 0.05 | 6 | 0.652 | 0.6733 | 13 | 0.4663 | 0.5578 | 18 | 0.4986 | 0.4669 |
| 0.1 | 6 | 0.6131 | 0.6437 | 13 | 0.4461 | 0.5060 | 18 | 0.4893 | 0.4157 |
| 10 | 6 | 0.4099 | 0.5133 | 13 | 0.4887 | 0.3733 | 18 | 0.5029 | 0.3847 |

TABLE S1. Measured exponents $\nu_1$ and $\nu_2$, characterising the power law behaviour of $R_e$ as a function of $N$ for tangentially active linear polymers at different values of Pe and $R$.

confinement $R_e^\perp$ has a constant value for all $N$ considered; indeed, taking $\tilde{N} = 200$ for consistency, the two exponents are compatible, which suggests that there is a single power law. For all other cases, the two power law exponents are considerably different. Interestingly, this breaks at high activity, where the saturation has not yet been reached at $\tilde{N} = 200$. However, two distinct power laws are visible. This suggests that, possibly, a further regime could appear at high activity for much larger values of $N$.

## S3.   RADIAL AND ANGULAR DISTRIBUTIONS

### A.   Radial distributions of the centre of mass

We report here additional data on the probability of finding the centre of mass of the polymer at a certain $r$, the radial coordinate of the channel. The longitudinal axis, at the centre of the channel, is taken as $r = 0$.

We report the data in Fig. S4. As in the main text we observe that, in strong confinement conditions or at weak activity, the accumulation at the boundary of the channel is reduced, as compared to Active Brownian Particles.

### B.   Radial distributions of the monomers

We discuss here the probability of finding a monomer at a certain distance $r$ from the channel axis.

We report the data in Fig. S5. This is a complementary, more microscopic measure with respect to the radial distribution of the centre of mass. Interestingly, we can observe that also monomers do not accumulate at the boundary in the weak activity or strong confinement cases. Instead, there is a slight tendency to accumulate in the weak confinement, strong activity limit.

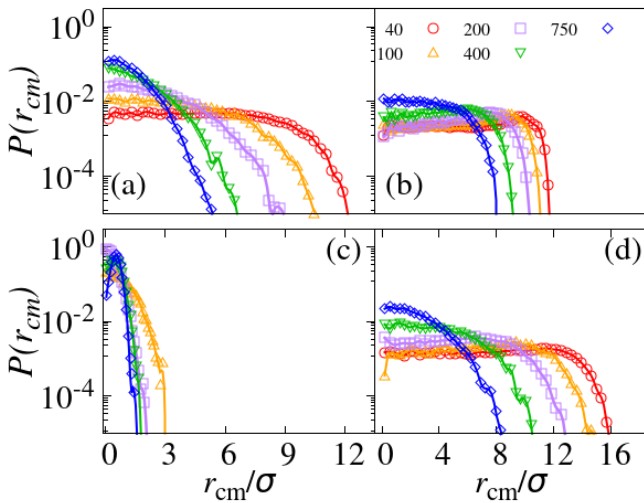

FIG. S4. Distribution of the position of the centre of mass of the polymers as a function of the radial distance from the channel centre for different values of $N$ and (a): $R = 13\sigma$ Pe $= 0.03$ (b): $R = 13\sigma$ Pe $= 10$ (c): $R = 6\sigma$, Pe $= 0.1$ (d) $R = 18\sigma$, Pe $= 0.1$.

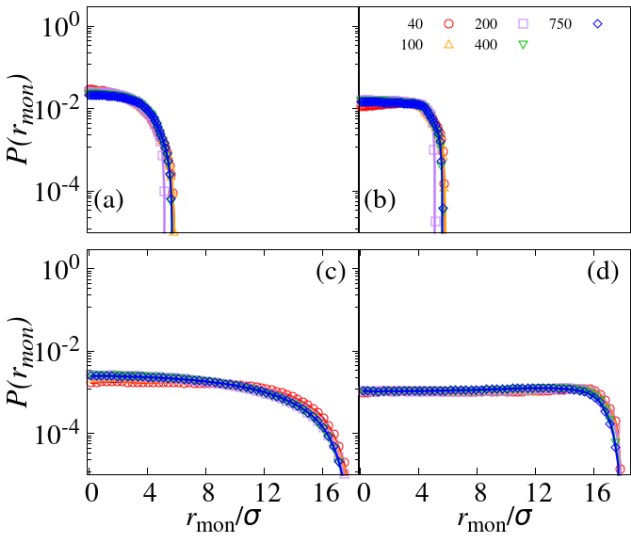

FIG. S5. Distribution of the position of the monomers as a function of the radial distance from the channel centre for different values of $N$ and (a): $R = 6\sigma$ Pe $= 0.03$ (b): $R = 6\sigma$ Pe $= 10$ (c): $R = 18\sigma$, Pe $= 0.03$ (d) $R = 18\sigma$, Pe $= 10$.

### C. Distribution of the relative orientations

We report here additional data on the distribution of the orientation of the end-to-end vector with respect to the channel axis. We report the data in Fig. S6. As also reported in the main text, we observe that alignment becomes important under strong confinement conditions. At intermediate confinement, the longest polymers will tend to align if Pe is small.

## S4. DIFFUSION COEFFICIENT FOR ACTIVE POLYMERS UNDER CYLINDRICAL CONFINEMENT.

We report the same data, reported in Fig.8 of the main text, plus additional data at different values of Pe (Pe $=$ $0.05, 0.1$) in a different fashion, to compare with the data reported in the literature in the bulk case[1]. Indeed, in Fig. S7 we report the Long time diffusion coefficient $D_a^C$, normalized by the diffusion coefficient of a single monomer $D_0$ as a function of Pe. In this way, we can observe that some of the characteristic features of these active polymers

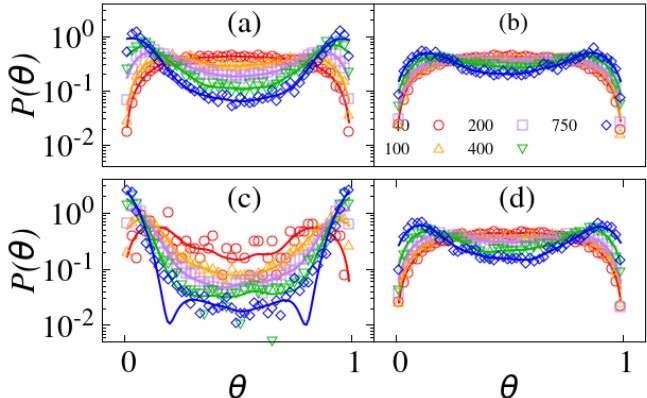

FIG. S6. Distribution of the angles between the instantaneous end-to-end vector and the channel axis for different values of $N$ and (a): $R = 13\sigma$ Pe = 0.03 (b): $R = 13\sigma$ Pe = 10 (c): $R = 6\sigma$, Pe = 0.1 (d) $R = 18\sigma$, Pe = 0.1.

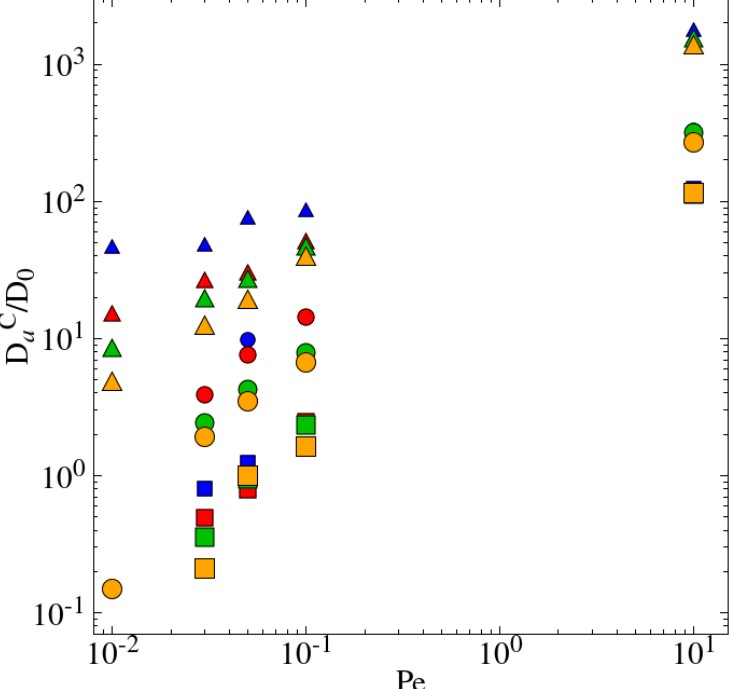

FIG. S7. Long time diffusion coefficient $D_a^C/D_0$ as a function of Pe for different values of polymer length ($N$ =40, squares; 100, circles; and 400, triangles) under different confinement conditions ($R$=6, blue; 10, red; 13, green; and 18, orange).

are maintained in confinement as well. Indeed, $D^C/D_0$ increases upon increasing Pe and, for large values of Pe and weak confinement conditions, the diffusion coefficient is roughly independent on $N$. However, the diffusion coefficient is observed to increase, upon increasing $N$ under strong confinement. This could be observed in the main text, as the data strongly deviate from the bulk ($N$-independent) trend.

[1] V. Bianco, E. Locatelli, and P. Malgaretti, Globulelike conformation and enhanced diffusion of active polymers, Physical Review Letters **121**, 217802 (2018).