# Peer review of "Tangentially Active Polymers in Cylindrical Channels"

_SciPost Physics_

## Round 1 · Referee Report · Anonymous (Referee 1) · 2024-6-10

Strengths
See my report attached.
Weaknesses
See my report attached.
Report
See my report attached.
Requested changes
See my report attached.
Recommendation
Publish (easily meets expectations and criteria for this Journal; among top 50%)
Author: José Martín on 2024-07-18 [id 4632]
(in reply to Report 1 on 2024-06-10)
""Strengths: The field of active polymers is a growing area of research that has recently received significant attention. This paper is undoubtedly timely; until now, the effects of boundaries and confinement on active polymers, unlike their passive polymer or active point-like counterparts, have not been properly considered and understood. As someone who is passionate about this field, I found the manuscript both pleasant to read and easy to follow. The paper is logically organized into distinct sections, providing enough detail for anyone interested in reproducing the results. The findings presented here are highly encouraging because they have solid implications for the development of a more general theory for active polymers and their collective behavior. Particularly, the authors show that the effect of confinement on active polymers reveals interesting new physics that is not intuitive. Specific Issues, Weaknesses, and Recommendations: I would be happy to see this paper published, but I have a few remarks - mostly minor points or requests for clarification - throughout the text and graphs that I would like to see addressed before recommending the final publication. In the following, I provide a chronological list of points (in text order) that will help the authors resolve these issues.""
We thank the Reviewer for positively acknowledging our work and for stressing that our results are timely. We provide, in what follows, a point-to-point answer to their comments.
""Abstract, page 1. "We observe that the scaling of the polymer size in the channel, quantified by the end-to-end distance, shows different anomalous behaviours at different confinement and activity conditions." Please change "at" to "under." Additionally, for clarity, the authors should mention that these "anomalous behaviours" are compared to those of their passive counterparts.""
We thank the Reviewer for this comment. We have included the proposed changes.
""Page 2. Figure 1 is currently just a cosmetic cartoon. I strongly recommend that the authors introduce the relevant notation (F a for the polymers and the boundary conditions (such as beads and size) in the sketch.""
We have included some of the proposed changes. Regarding the confining beads, we have chosen to report them in the snapshots, placed in the figures, as the result of including them in the sketch was not satisfactory. We have added a short comment in the caption.
""Page 2. Can the authors comment on the role of the size of the beads in the boundary conditions? Since the characteristic size of the bead, $\sigma$, is chosen to be the same as for the beads in the boundary conditions, can the chain be trapped or "feel" the voids between the beads of the wall?""
We thank the Reviewer for this comment. As already hinted in the answer to another Reviewer's comments, we employed such a methodology to be consistent with a follow-up work on corrugated channels, in which having a smooth wall would be challenging. We actually did some preliminary tests in the more complex geometry of the corrugated channels, by setting confining walls with more beads (so, smoother walls). The tests reassured us that there was little change in the results and, for computational reasons, we choose to employ the strategy described in the main text. The roughness is effectively (at most) $\sigma/2$: in other words, the "voids" between the channel beads are smaller than the monomer size. Being the channel steric (i.e. only repulsive), its roughness may have a quantitative influence on the results, but the polymer can not be trapped on the inner surface.
""Page 2. Can they also provide a more in-depth explanation for the specific choice of $\epsilon$ = 10, $k_B T$ and $K = 30 \epsilon$ for their simulation? They briefly mention a rather vague explanation: "This choice of parameters avoids strand crossings." Please elaborate on the rationale behind this particular choice.""
We thank the Reviewer for this interesting comment. As the name suggests, bead-spring models represents the polymers as a collection of beads (coordinates in a Cartesian frame) and springs (pair potentials). In general, neighbouring monomers have a short-range repulsion and a long-range attraction, with a minimum at the equilibrium (rest) length. In our case, we use a combination of WCA and FENE potentials. Bonds are "stiff" if very large forces (compared to the thermal ones) are required to deviate from the equilibrium length. Self-avoiding polymers have an additional short range repulsion (WCA again in our case) for non-bonded beads, i.e. for beads that are not part of the same bond. Strand crossing, in bead-spring models, happens when a bond stretches so much that another bonded pair (at some distance along the contour) can cross it. For polymers modeled with the WCA+FENE, the strands can not cross because of the the short range repulsion between the beads belonging to the two different bonds. However, even in equilibrium, stretching may be caused by the combination of thermal fluctuations and (especially at high density) steric forces: this event is extremely unlikely due to the bond stiffness (one may check the original work of Kremer and Grest Ref.[67]). The presence of tangential forces makes this event, anecdotally, more prevalent, even at infinite dilution. Strand crossing creates nonphysical transitions between conformations and affects the gyration radius of the polymer, as well as other conformational and dynamical properties, even in equilibrium. As such, we strove to avoid it. An effective solution to the problem consists in increasing the value of the parameter $\epsilon$, that sets the potential energy scale in relation to $k_B T$. By increasing $\epsilon$ (and, as a consequence, the parameter $K$ in the FENE potential), one stiffens the spring, i.e. larger forces are required to deviate from the equilibrium length. Notice that the equilibrium length does not depend on $\epsilon$ and so it is left unchanged by this particular strategy. We added a comment in the text about the chosed value: "... where we set $K=$30$\epsilon /\sigma^2$=300$k_B T/\sigma^2$ and $R_0=1.5 \sigma$. This choice of parameters allows to avoid strand crossings that could be relevant in the measured properties even in the passive case (Kremer and Grest, JCP, 92, 5057 (1990))."
""Page 3, Part C. It took me a long time to figure out what the different exponents and subscripts meant. I think it would be very helpful for any reader to explicitly spell out in the text what they are standing for (e.g.,C for confinement, etc.).""
We thank the Reviewer for pointing out this issue. We have revised the manuscript thoroughly in order to clean up the presentation. We have added the following text at the beginning of section 3: "In what follows, we will indicate quantities with a superscript $C$ and $B$ for confinement and bulk, respectively; quantities that refer to the passive limit will be also tagged with a $p$ and we will omit the equivalent tag ``$a$'' in the active case, for the sake of simplicity.""
However, we did not highlight all the notation changes.
""Page 3. In equation (10), I believe that $\gamma$ should be read as the monomer friction coefficient $\gamma_0$.""
We thank the Reviewer for spotting this. We have revised and corrected the notation.
""Page 4, Equation (17). Please define the quantities in the text.""
We thank the Reviewer for this issue. We have added that the parameters in the equations are fitting parameters. The quantities $N$ and $\sigma$ were already defined.
""Page 5, Figure 2. I found the graph difficult to follow, and thus poorly convincing, at first glance, even with the help of the text. Initially, it is challenging to appreciate the significance of the changes induced by activity and for different confinements; although the different scalings are locally indicated, understanding what is interesting here proved difficult.\\
(i) I suggest that the authors first indicate the Péclet (Pe) values in (a) and (b) to guide the reader on what they are observing (essentially low activity versus high activity). I am aware that this information is provided in the caption, but the numerous variables make it hard to follow. Additionally, could they highlight the ``cryptic'' value N = 200 in this graph with a vertical line and elaborate on this critical value? Why is this value significant?\\
(ii) Since one of the main claims of the paper is that even low activity
levels (as low as Pe = 0.03) affect the conformation of the driven polymer in confinement, it is essential to show the conformation of a purely passive polymer in this graph.""
We thank the Reviewer for pointing out this issue. We have revised the presentation of Fig.~2, essentially plotting less data and adding a comparison between the passive end-to-end distance under confinement and the active end-to-end distance in the bulk. In the text, we elaborate more on the changes to the power law scaling exponents, comparing directly with the passive confined and the active bulk cases. We have also rewritten the text on $R_e^{\perp}$ and interpreted the quantity more carefully. As a result of this analysis, $\tilde{N}$ does not have a constant value and simply marks the onset of the constant plateau. That said, at least for these data, we have kept the estimate for the exponents $\nu_1$ and $\nu_2$, as $\tilde{N}=200$ is still a very good, practical condition to discriminate between the regimes emerging from the data.
""Page 6, figure 3), Please also indicate the Pe values in the graph.""
We have added the values of $\mathrm{Pe}$ in the graph. The figure has also been revised.
""I strongly recommend that the authors add snapshots (none is shown!) of the conformation of the driven polymer in the figures when an effect is highlighted or discussed. Alternatively, they could include these snapshots as an independent figure. This addition would greatly improve reader understanding.""
We thank the Reviewer for this suggestion. We added snapshots throughout the manuscript.
""Page 6, Figure 4. I have the same comment as before regarding the Péclet (Pe) values. Additionally, please indicate in the caption or legend what the continuous line represents (e.g., equation 17).""
We have added the values of $\mathrm{Pe}$ in the graph and we have updated the caption.
""Page 6, Figure 5. Could the authors comment on the non-monotonous variation of the probability density function for N = 750, low Pe?""
We thank the Reviewer for this question. We do not have a clear explanation of this phenomenon. We have verified that the data for active polymers is sound, as we are computing our statistics over trajectories that are much longer than the longest correlation time of the polymer and over $M=200$ independent realisations. The mentioned issue is very interesting and we will most definitely try to investigate the emergence of this non-monotonicity in the future.
""Page 7, Figure 6. Why not use symmetry (i.e., only show half of the axis)?""
We thank the Reviewer for this suggestion. We have reviewed the figure as suggested.
""Page 7, Figure 7. I appreciated the results presented here and the ensuing discussion. However, I still feel a comment on the critical value of N/Pe, at which a switch between the bulk scaling and the other scaling is observed, would be beneficial. Additionally, it would be necessary to present this graph without rescaling to appreciate the effectiveness of the proposed rescaling.""
We thank the Reviewer for this suggestion. We have tried with different fittings and dependencies for the reorientational time. However we couldn't find a way to properly observe the cross-over. More work will be needed to address this issue, but it goes beyond the scope of this manuscript. We plan to include such study in a fore coming manuscript. We have now included a figure for the reorientational time as a function of the polymer size without any rescaling in the Supplementary Material.
""Page 8, Figure 8. For clarity, add labels such as "high confinement" in (a) and "low confinement" in (b).\\
(i) My main question pertains to the conformations here. One could show
some snapshots as a function of Pe (from blue to red) which would, in my
opinion, better rationalize the findings in (a).\\
(ii) Moving from low Pe values (blue) to high Pe values (red), the behavior of the chain follows the bulk behavior more closely. From the text, it is not really clear what is happening here. Higher activity generally means stiffer behavior (Re increases) for tangentially driven chains. Does this imply that active forces are more aligned and thus the chain is propelled faster? Why would it deviate under higher confinement? What is the physical picture?\\
(iii) At low confinement (b) and low activities (0.03 and 0.05): why is there a deviation from the scaling arguments for shorter chains?""
We thank the Reviewer for these suggestions. We have added the suggested lables in the panels. Regarding the rest
(i) We have added a few snapshots.
(ii) Upon increasing $\mathrm{Pe}$, the polymers assume a more compact conformation, that is indeed more bulk-like. As such, they are less affected by the confinement and, overall, less aligned with the channel axis (as Fig. 6 shows). Our model rationalizes this through the geometrical construction of $\theta_{\mathrm{max}}$. We have clarified the text, adding a short mention on the crucial role of the alignment.
(iii) For $\mathrm{Pe} \ll 1$ and for small chains, $R_e^C$ becomes a bit smaller than $R_e^B$. This happens also in passive polymers, for some values of the ratio between the confined end-to-end distance and the channel radius. This effect is certainly amplified by the activity; the discrepancy can be appreciated in Fig.~4. Since for small chains there is no alignment effect and no difference in the correlation time with respect to the bulk, the difference is just in the polymer extension. We have added a comment in the text.
""Page 9 "large aspect rations"- typo - change to ratio""
We thank the Reviewer for spotting this typo. We fixed it.
Anonymous on 2024-05-27 [id 4519]
In this interesting manuscript authors consider a numerical simulation of an active polymer. Authors remark "We perform Langevin Dynamics simulations, in the overdamped regime, disregarding hydrodynamics. We employ the open source package LAMMPS, with in-house modifications to implement the tangential activity."
Unfortunately, even though an open source package, with an open license (GPL 2.0) is chosen as a starting point authors do not disclose the code used in their simulations. 1. Such choice makes this numerical-simulation-based manuscript very difficult and perhaps impossible to reproduce / validate. 2. Performing modifications/enhancements of open packages without sharing with wider scientific community goes against principles of open access central to SciPost's mission 3. GPL 2.0 license explicitly requires modified programs to be made available under the same license (or not at all)
I hope this serious oversight is easy to address and high quality, well documented source code repository will accompany this numerics-focussed manuscript. This would also allow for further improvements of simulation method. The authors choose Velocity Verlet algorithm which is not the best choice for simulations where nearest neighbours along the chain are connected using stiff springs. The choice of Velocity Verlet is even stranger given that LAMMPS already implements rRESPA family of algorithms to deal exactly with this type of problem.

---

## Round 1 · Referee Report · Anonymous (Referee 3) · 2024-6-20

Strengths
1) Interesting paper, very thorough in the data analysis and gathering
2) New nontrivial results in the field of active polymers under confinement
Weaknesses
1) Figures can be improved, they are at times hard to read and interpret
2) I have a number of points that need to be addressed
Report
This is an interesting paper. The authors consider the problem of a tangentially driven active polymer under cylindrical confinement.
They study the system under three different degrees of confinement for for two strength of the active forces. They also map their system to that of a single Brownian active particle and study structural and dynamic properties of the system.
They find a couple of intriguing results that are notable and worth of publication, including
1) The way active polymers accumulate on the confining surface, is quite different than how single particles behave near them.
2) This papaer show how the blob scaling used for passive polymers breaks down
for tangential active polymers, and that radius of the channel is not a fundamental length scale of this problem.
Requested changes
1) The conclusions could be written more clearly.
2) What is the interaction between the beads defining the channel and the monomers?
3) In Figure 2 it would be very useful to mark in the horizontal axis at which size N, the passive radius of gyration of the polymer becomes comparable to the degree of confinement, just to have a sense of the degree of confinement the reference passive system would be experiencing.
3) The authors say
“Indeed, here \nu is reminiscent of the passive exponent under confinement (\nu =1); however, as the tangential activity tends to shrink the polymer chains, the final outcome results from their interplay.”
The \nu=1 is only true under very strong confinement. For a passive polymer under weak confinement (small N) that should be ~0.6. I am not sure the authors are in the limit of very strong confinement, as even for the longest polymer discussed in Fig2 (N=200) I would expect a radius of gyration of the order of ~ 15
This is confusing. I get that the activity reduce the power law dependence of Re, I am not sure the reference limit is correct here.
4)
Put more ticks in the x axis in Figure 3.
The saturation of the red-points in the two figures does not seem to happen at the same value of N as discussed in the paper. Add horizontal lines (or colored dots on the y-axis) to indicate the size of the confinement (R).
4) In discussing Fig. 4 the authors say:
“We observe that, in some regimes of confinement and activity, sufficiently small active polymers behave as their passive counterpart (gray line)”
There is no gray line in the figure (should that be black?).
5) The authors say:
“More importantly, given this rescaling, the reported data do not collapse on a single universal curve.” However the data in Fig.4 (b) seems to suggest that they do in the large Pe limit.
What am I missing? That seems like a pretty good collapse to me, it does not follow the passive prediction, but the data seem to collapse into a master curve nonetheless.
Recommendation
Ask for minor revision
Author: José Martín on 2024-07-18 [id 4634]
(in reply to Report 3 on 2024-06-20)
""This is an interesting paper. The authors consider the problem of a tangentially driven active polymer under cylindrical confinement. They study the system under three different degrees of confinement for for two strength of the active forces. They also map their system to that of a single Brownian active particle and study structural and dynamic properties of the system. They find a couple of intriguing results that are notable and worth of publication, including 1) The way active polymers accumulate on the confining surface, is quite different than how single particles behave near them. 2) This papaer show how the blob scaling used for passive polymers breaks down for tangential active polymers, and that radius of the channel is not a fundamental length scale of this problem.""
We thank the Reviewer for their appreciation of our work and, in particular, for stressing that our results are notable and worth of publication. We provide, in what follows, a point-to-point answer to their comments.
""The conclusions could be written more clearly.""
We have revised the conclusions, taking into account the other Reviewer's comments.
""What is the interaction between the beads defining the channel and the monomers?""
We thank the Reviewer for this question. We employed the WCA potential to model the repulsion between the immobile beads, that constitute the confinement, and the monomers. We have added the information in the main text: "The interaction between the beads and the monomers is given by the WCA potential, Eq. (1)"
""In Figure 2 it would be very useful to mark in the horizontal axis at which size N, the passive radius of gyration of the polymer becomes comparable to the degree of confinement, just to have a sense of the degree of confinement the reference passive system would be experiencing.""
We thank the Reviewer for this comment. We have completely re-plotted Fig.2 and now we bring forward a direct comparison with the passive case. This curve essentially includes the information requested by the Reviewer: as an example, the end-to-end distance under confinement reported as a function of $N$ clearly changes power law behaviour upon reaching the strong confinement condition.
""The authors say ``Indeed, here $\nu$ is reminiscent of the passive exponent under confinement ($\nu$ =1); however, as the tangential activity tends to shrink the polymer chains, the final outcome results from their interplay. The $\nu$=1 is only true under very strong confinement. For a passive polymer under weak confinement (small N) that should be ~0.6. I am not sure the authors are in the limit of very strong confinement, as even for the longest polymer discussed in Fig2 (N=200) I would expect a radius of gyration of the order of ~ 15. This is confusing. I get that the activity reduce the power law dependence of Re, I am not sure the reference limit is correct here.""
We thank the Reviewer for this comment. We have revised the presentation and updated Fig.~2. In brief, we want to highlight that both the presence of confinement and the tangential activity affect the polymer conformation and, as such, the final result will depend on their interplay. In the new Fig.~2, we show that, at fixed $R$ and for small values of $\mathrm{Pe}$, small polymers behave as passive. However the data show that the effects of the interplay are visible for larger polymers, even for $\mathrm{Pe} \sim 10^{-2}$ (notice that the largest polymer simulated here is $N=$750, not $N=$200). Indeed, focusing here on $R=6$, passive polymers with $N\approx$ 100 are strongly affected by the confinement. So, the interplay of the elongation caused by the confinement and the compaction caused by activity results in a non-trivial exponent at large values of $N$. For example, we get $\nu_a^{C}\approx$0.67 at $\mathrm{Pe}=$0.03, that is larger than the one expected for active polymers in bulk (e.g. $\nu_a^{B} \approx 0.56$ at the same value of $\mathrm{Pe}$) but also smaller than for passive polymers under confinement ($\nu_p^{C}=$1). Other cases are discussed in the main text.
""Put more ticks in the x-axis in Figure 3. The saturation of the red-points in the two figures does not seem to happen at the same value of N as discussed in the paper. Add horizontal lines (or colored dots on the y-axis) to indicate the size of the confinement (R).""
We thank the Reviewer for this comment. We have modified the figure, plotting $R_e^{\perp}/R$ and including horizontal lines for the maximum value of the end-to-end vector. We have modified the text accordingly. We believe that this way of plotting the data has the advantage of better highlighting the value of $\tilde{N}$, that is indeed not constant.
""In discussing Fig. 4 the authors say: "We observe that, in some regimes of confinement and activity, sufficiently small active polymers behave as their passive counterpart (gray line)'' There is no gray line in the figure (should that be black?)""
We thank the Reviewer for spotting this typo, that has now been corrected.
""The authors say: "More importantly, given this rescaling, the reported data do not collapse on a single universal curve." However the data in Fig.4 (b) seems to suggest that they do in the large Pe limit. What am I missing? That seems like a pretty good collapse to me, it does not follow the passive prediction, but the data seem to collapse into a master curve nonetheless.""
We thank the Reviewer for this comment. Indeed, in general the data show a good collapse, within the chosen range of values. However, in figure 4 it is possible to observe that the collapse is not always good when comparing results obtained for the active polymer to those for the passive case (black curve in figure 4). Data reported in panel a-figure 4 (low Peclet) present a worse collapse than data in panel b-figure 4 (large Peclet). Unfortunately, at the moment we do not have a clear and strong argument for justifying this scaling. Work is in progress in this respect, and will be the subject of a future manuscript.

---

## Round 1 · Referee Report · Anonymous (Referee 2) · 2024-6-20

Strengths
1 - An interesting problem in the physics of active matter systems
2- Very clean and carefully done numerical simulations
3- Extensive and careful analysis of the results
Weaknesses
1 - Limited relevance to biological systems
2- A bit artificial model of activity
Report
Overall, this is quite interesting and carefully done numerical work with good comparison to theory. The problem is of interest to the community working on active matter systems, in particular those working on problems that go beyond simple active agent models. The model is, however, a bit artificial in terms of the way the activity is introduced - most biological filaments involve some molecular motor that acts to slide pairs of filaments against each other. So, it is not clear how relevant the results would be to biology.
I am also wondering how sensitive are the results on the way the confinement is implemented. At present, the authors use a set of fixed beads which might introduce roughness to the wall. Implementing the actual confinement constraint is conceptually not hard, but might be technically challenging in LAMMPS. Could they please comment?
Requested changes
1 - Please add legends to most figures
2 - Please comment in the caption of Fig. 4 about the solid black line
3- Please make notation consistent between that main text and the SI (e.g. vectors in the main text are shown in bold, and in SI they have overhead arrows).
4 - Full-line equations are part of the text and should include proper punctuation.
5- The last paragraph in Conclusions is highly speculative and vague. Please either remove it or make it more precise.
6 - Please be consistent with the space between a word and the citation that follows it.
Recommendation
Publish (easily meets expectations and criteria for this Journal; among top 50%)
Author: José Martín on 2024-07-18 [id 4633]
(in reply to Report 2 on 2024-06-20)
""Overall, this is quite interesting and carefully done numerical work with good comparison to theory. The problem is of interest to the community working on active matter systems, in particular those working on problems that go beyond simple active agent models. The model is, however, a bit artificial in terms of the way the activity is introduced - most biological filaments involve some molecular motor that acts to slide pairs of filaments against each other. So, it is not clear how relevant the results would be to biology.""
We thank the Reviewer for their positive remarks. We provide, in what follows, a point-to-point answer to their comments. Regarding the way the active propulsion is introduced, few systems exist in Nature where single filaments are propelled. Most notable examples are motility assays of microtubules on carpets of molecular motors, RNA and DNA under the action of polymerases, cyanobacteria and similar micro-organisms, FtZ filaments and also worms. Not all of these systems can be described exactly by a tangential propulsion, but none presents a sliding mechanisms, such as in actomyosin bundles or in some experimental realisations of active nematics. We also remark that, in this work, we are not trying to model a particular biological system, but rather to establish generic features for tangentially self-propelled filaments under cylindrical confinement. As mentioned in the conclusions, this could be interesting for worms and filamentous bacteria.
""I am also wondering how sensitive are the results on the way the confinement is implemented. At present, the authors use a set of fixed beads which might introduce roughness to the wall. Implementing the actual confinement constraint is conceptually not hard, but might be technically challenging in LAMMPS. Could they please comment?""
We do not expect any qualitative influence of the implementation of the confinement on the observed properties. We used a rough surface to be consistent with a follow-up work on corrugated channels, in which having a smooth wall would be challenging. We have included a reference in the main text to this paper for the interested reader. We actually did some preliminary tests in the more complex geometry of the corrugated channels, by setting confining walls with more beads (so, smoother walls). The tests reassured us that there was little change in the results and we choose to employ the strategy described in the main text. Finally, the roughness is effectively (at most) $\sigma/2$: being the channel steric (i.e. only repulsive), its roughness may, in general, have a quantitative influence on the results, but the polymer can not be trapped on the inner surface.
""Please add legends to most figures""
We thank the Reviewer for this comment. We have modified the figures as suggested.
""Please comment in the caption of Fig. 4 about the solid black line""
We thank the Reviewer for this comment. We have added the description of the black line in the caption.
""Please make notation consistent between that main text and the SI (e.g. vectors in the main text are shown in bold, and in SI they have overhead arrows).""
We have modified the text as suggested. Now all vectors are in bold font.
""Full-line equations are part of the text and should include proper punctuation.""
We have modified the equations as suggested.
""The last paragraph in Conclusions is highly speculative and vague. Please either remove it or make it more precise.""
We thank the Reviewer for this comment. We have removed the last paragraph.
""Please be consistent with the space between a word and the citation that follows it.""
We have modified the text as suggested.

---

## Editorial Decision

unknown